# Psycho-social problems among older people residing in community of Chitwan, Nepal—A cross-sectional study

**Sunita Poudyal**[ID]*, **Kalpana Sharma, Hem Kumari Subba, Ramesh Subba**

School of Nursing, Chitwan Medical College, Bharatpur, Nepal

* poudyal.sunita@cmc.edu.np

**Data Availability Statement:** All relevant data are within the manuscript and data set are also deposed directly to. OSF registries Registration DOI https://doi.org/10.17605/OSF.IO/HTGYD.

## Abstract

### Background

Older people are vulnerable to various psycho-social problems such as depression, anxiety, insomnia, dementia, and loneliness that have profound impact on overall health and their quality of life and higher use of health services. Prevention and treatment of psychological problems in this risk group is critical for improving their quality of life.

### Objective

This study aimed to find out the psycho-social problems among older people residing in community.

### Methods

The study was a cross-sectional study design and 388 older people aged 65 years and above residing in different wards of Bharatpur Metropolitan city were selected using probability simple random sampling technique. Data were collected using interview schedule containing geriatric depression scale-15; Geriatric anxiety scale-10, University of California Loss Angels Loneliness Scale (UCLA-20), Anthens Insomnia Scale and Mini-Cog Test. Obtained data were analyzed in SPSS version 20 for windows. Chi-square test was applied to measure the association between psychosocial problems and selected variables.

### Result

Findings of the study revealed that the mean age (± SD) of respondents was 72.92 (±7.12) years. Almost all (93.6%) older people had full functioning of activity of daily living and two third (66.5%) had other co-morbid conditions. More than two third (67.0%) of older adults had depression, 60.3% had anxiety, 53.6% had moderate to high level loneliness, 47.2% had insomnia, and 33.3% had dementia. Age, functional dependency, sex, co-morbidity, financial dependence, education and occupation were significantly associated with the psychosocial problems among older people.

**Funding:** This study is funded by Nepal Health Research Council as Provincial Health Research Grant FY 2079/080. But the funder had no role in study design, data collection and analysis, decision to publish and preparation of manuscript.

**Competing interests:** The authors have declared that no competing interests exist.

## Conclusion and recommendation

Psychosocial problems are common among older people residing in community of Chitwan. Hence, there is need to develop and implement health care strategy by local health care planner to prevent, treat and manage the psychosocial problems among this risk groups. Further, health care providers working in geriatric problems or psychosocial health need to conduct regular screening programs for the early diagnosis and treatment of these problems.

## Introduction

The number of older people population is increasing dramatically around the world, including in Nepal. In recent decades, the global population of older adults has grown steadily and proportionally. In 2019, the number of persons aged 60 and over was 1 billion, which will continue to rise to 1.4 billion by 2030 and 2.1 billion by 2050 [1]. The growth rate of older adults in developing countries is much higher than that of developed countries [2]. In Nepal, there had 2.1 million senior people, accounting for 8.1% of the country's total population [3].

Population aging is inevitable and a positive aspect of human development [4]. However, it poses a number of problems among older people. Older people are more likely to suffer from chronic or persistent somatic or physical illnesses, such as hearing loss, diabetes, cancer, arthritis, cardiovascular and/or respiratory conditions and respiratory disorders [5–8]. They face age related difficulties such as diminished mobility, persistent pain, fragility and unpleasant life experiences, bereavement and financial dependency etc. These all may lead to psychosocial problems [5–9] such as social isolation, loneliness, or emotional distress, anxiety, depression, poor relationships, financial constraints, and a lack of social networks [9–11].

Evidence showed that older people suffer from different psycho-social problems such as depression, dementia, agitation, anxiety, loneliness, social exclusion and burden to their family which have subsequent effects on their physical health [12–14]. Dementia and depression are the most prevalent illnesses affecting 5% and 7% of the senior citizens globally, respectively. Dementia is a major contributor to impairment and dependency among older people worldwide, ranking seventh among all diseases in terms of cause of death [1]. Similarly, anxiety disorders account for around 25% of fatal self-harm incidents in older adults, affecting 3.8% of them [9].

Older people residing in Nepal are not far from psychosocial and physical health problems. The prevalence of depressive and anxiety symptoms ranged from 4.4% (in community) to 53.2% (in hospital), and 21.7% to 32.3% respectively [15]. Similarly, 75.65% of senior citizens in geriatric homes had dementia symptoms [16]. Further, evidences showed that the severe psychosocial problems are higher in institutionalized older people compared to living in a family [17, 18].

Psychosocial problems are common among the older population but these conditions are usually under diagnosed and undertreated due to their complex and multi factorial nature. In addition, there is a dearth of information regarding these issues in our context. Hence, this study aimed to identify the psycho-social problems among older people residing in community which help to bridge the evidence gap for the successful implementation of program regarding these issues.

## Materials and methods

### Study design and setting

A cross-sectional survey was carried out in Bharatpur Metropolitan City, Chitwan. The Chitwan district consists of seven municipalities. Out of which one is a metropolitan city, five are urban municipalities, and one is a rural municipality. Bharatpur metropolitan city is one of the fastest-growing cities in Nepal, consisting of 29 wards.

### Study population and eligibility criteria

The population of the study were those people who were 65 years and older and residing in community of Bharatpur, Chitwan. Those people who were able to give consent and willing to participate in the study were included whereas those old people who were already diagnosed with mental illness and/or were chronically ill and unable to communicate were excluded from the study.

### Sample size determination and sampling technique

Cochrane's formula ($n_0 = z^2pq/e^2$) was used to calculate the needed sample size for the study considering a confidence level of 95%, an allowable error of 5%, and 45.6% prevalence of various psycho-social problems among older people in India [19].There were total 4570 older people, where in ward no. 5 = 938, ward no. 10 = 1707, ward no. 12 = 961 and ward no. 27 = 964 [20]. Calculated sample size was 352. Adding 10% non-response error rate, final sample size (n) was 388.

Two-stage sampling techniques were applied. Initially, 4 wards (ward no 5, 10, 12, and 27) were selected randomly from out of 29 wards in Bharatpur Metropolitan City. Second, a list of the older adults residing in those selected wards was obtained. Third, the desired sample was selected using probability proportionate to size.

### Data collection tools and measurement

The structured interview schedule was used to collect socio-demographic information and health related information. Katz Index of Independence in Activities of Daily Living [21] was used for the assessment of adequacy of performance in the six functions of bathing, dressing, toileting, transferring, continence, and feeding. A score of 4 to 6 indicates full function, 2.1 to 4 indicate moderate impairment, and 2 or less indicates severe functional impairment. Smoking status was classified as never smoker (who had never smoked) ex-smoker (who ever smoked any tobacco products in his or her lifetime but has not smoked in the last 28 days) and current smoker (who smoked currently any tobacco products such as cigarettes, pipes or cigars). Similarly, alcohol users were categorized as lifetime abstainer (never consumed alcohol in their lifetime), former drinker (used to drink alcohol but have abstained for the last 12 months) and current drinker (consumed alcohol at least once during the last 12 months and who currently drink).

For the assessment of psycho-social problems among older people, five types of tools were used: Geriatric Depression Scale-15 (GDS-15) [22] for the assessment of depression, Geriatric Anxiety Scale –10 (GAS-10) [23] for anxiety, Revised UCLA Loneliness Scale [24] for the loneliness and social isolation, Anthens Insomnia Scale (AIS) [25] for the prevalence of insomnia, and Mini-Cog Test [26], a brief cognitive test for the assessment of dementia.

GDS-15 is a valid and reliable screening tool for geriatric depression with the Cronbach alpha coefficient value 0.920 [27]. It is also validated in Nepalese context with Cronbach's alpha value 0.79 [28]. It consisted of 15 questions and the response for these questions was yes,

or no. Total score ranged from 0 to 15. Scores of 0–4 are considered normal, 5–8 indicate mild depression; 9–11 indicate moderate depression and 12–15 indicate severe depression. GAS-10 [23] is a self-report measure of anxiety which had strong psychometric properties among older adults and used in Nepal [29]. It has 10 items scale ranged from 0 to 3 (0- Not at all, 1- Sometimes, 2- Most of the time and 3- All of the time). Items 1 through 10 were summed to provide a total score. The total score was calculated and classified as: Minimal anxiety ($\leq$6); Mild (7–9); Moderate (10–12); and Severe anxiety (>12). Revised UCLA Loneliness Scale [24] is highly reliable, both in terms of internal consistency (coefficient alpha = .96) and test-retest reliability (r = .73). Its reliability was tested in Nepalese young senior citizens with Cronbach's alpha 0.94 [30]. There were total 20 items including 10 positively worded and 10 negatively worded items designed to measure one's subjective feelings of loneliness as well as feelings of social isolation. Each item on a scale ranged from 1 (never) to 4 (often).The range of potential scores was 20 to 80. Negative items scores were reversed. The total score of loneliness was calculated and categorized as: Low (20–34), moderate (35–49), moderately high (50–64) and high (65–80). Anthens Insomnia Scale (AIS) is valid and reliable instrument with Cronbach's alpha value was 0.90 [25]. It consisted of eight sleep factors such as sleep induction, awakening during the night, final awakening earlier than desired, total sleep duration, overall quality of sleep, sense of well-being during the day, functioning (physical and mental during the day) and sleepiness during the day. Each item was rated on a 0–3 scale. Total score was calculated and further classified into two groups as insomnia ($\geq$6) and normal or no insomnia (<6). The Mini-Cog, is a rapid, valid and reliable screening tool for cognitive impairment [31]. Mini-Cog Test [26] assesses the older person's ability to recall three words and draw a clock. The recall test was graded on a scale of 0 to 3 and Clock Drawing Test (CDT) scored 0 or 2. The total score was calculated and classified as score of 0–2 dementia and score of 3–5 No dementia.

After obtaining data collection permission, researchers went to selected wards. Selected participants were identified with the help of social workers/community health volunteers. Objective of the study was explained to them and written informed consent was obtained. Those participants who gave consent were recruited in the study. Information related to socio-demographic, health related information and psychosocial problems were collected from the participants through face to face interview method using Nepalese version instruments. Each interview was taken 40–45 minutes and interview was done in separate room or corner of house with convenience of participants. The data were collected in two months period from 15th March 2023 to 15th May 2023.

**Data quality control.** The questionnaire were translated into Nepali language and back translated to English language by language expert for the consistency. Pre-testing of the instrument was done among older adult of similar characteristics residing in ward number 7 and excluded from the study. Reliability of the instrument was tested using data obtained from pre-test. Cronbach's alpha value >0.75 was considered as valid instrument for the final study.

**Data management and analysis.** Collected data were entered into Epi data 3.1 and exported to IBM SPSS (Statistical Package for Social Sciences) version 20.0 for window. Descriptive statistics were applied to report outcome variables and categorical variables. Chi-square test was performed to measure the association between selected variables and psychosocial problems. Statistical significance was set at p value <0.05.

## Ethical consideration

Before data collection, ethical clearance was obtained from Chitwan Medical College, Institutional Review Committee (CMC-IRC, Ref no- CMC-IRC/079/080/095). Data collection permission was obtained from each ward office of 5, 7, 10 and 12, Bharatpur Municipality.

Participants were informed about the purpose of the study. The written informed consent was obtained from all participants to maintain their right to self-determination. For the anonymity and confidentiality of information, code number was given to each participant instead of name, and they were assured that their information will not be disclosed to other unauthorized persons. Privacy was maintained during data collection by interviewing them in room or corner of house.

## Results

Majority of the old people were young old (61.9%), male (52.6%), followed Hindu religion (92.3%), lived in joint family (81.4%), and married and living with spouse (73.5%). Similarly, more than half (51.8%) were literate; majorities were farmer and home maker (70.1%) and were involved in work currently (72.7%). Few older people had got pension (15.2%) and two third (66.5%) of the respondents had co-morbid conditions. Similarly, few people were current smoker (12.4%) and current drinker (2.3%). Almost all (93.6%) had full functioning activity of daily living (Table 1).

Out of 388, 12.1% of older people had severe depression, less than half (47.2%) had insomnia, 1.5% had felt severe loneliness whereas more than half (54.6%) of older people felt moderate level of loneliness. Likewise, 33.5% of had dementia and 29.4% of had severe anxiety (Table 2).

Educational status and pension facility were significantly associated with the depression among older people. However, none of variables were significantly associated with the anxiety (Table 3).

The presence of co-morbidity and activity of daily living were significantly associated with the insomnia. Likewise, education was significantly associated with the loneliness among older people (Table 4).

**Table 1. Socio-demographic and personal characteristics of respondents n = 388.**

| Variables | Number (%) | Variables | Number (%) |
|---|---|---|---|
| **Age groups in years** | | **Currently working** | |
| Young old (65–74) | 240 (61.9) | No | 106 (27.7) |
| Middle old (75–84) | 121(31.2) | Yes | 282(72.7) |
| Old old (85 and above) | 27 (7.0) | **Pension facility** | |
| Mean ± SD = 72.92 (± 7.12) year Min age:65year Max age:99 year | | No | 329(84.8) |
| **Sex** | | Yes | 59 (15.2) |
| Male | 204(52.6) | **Co-morbidity** | |
| Female | 184(47.4) | No | 130(33.5) |
| **Family type** | | Yes | 258(66.5) |
| Nuclear | 72 (18.6) | **Activity of daily living** | |
| Joint | 316 (81.4) | Impaired | 25(6.4) |
| **Marital status** | | Full functioning | 363(93.6) |
| Married and living with spouse | 285 (73.5) | **Smoking habit** | |
| Single (widow, widower) | 103(26.5) | Never smoker | 210 (54.1) |
| **Educational status** | | Past smoker | 130(33.5) |
| Illiterate | 187(48.2) | Current smoker | 48 (12.4) |
| Literate | 201(51.8) | **Alcohol habit** | |
| **Previous occupation** | | Abstainer | 353(91.0) |
| Farmer/ Homemaker | 272(70.1) | Former drinker | 26 (6.7) |
| Service/ Business/ Labour | 116 (29.8) | Current drinker | 9 (2.3) |

**Table 2. Psychosocial problems among older people n = 388.**

| Psychosocial Problems | Number | Percent |
|---|---|---|
| **Depression** | | |
| Normal (0–4) | 128 | 33.0 |
| Mild (5–8) | 148 | 38.1 |
| Moderate (9–11) | 65 | 16.8 |
| Severe (12–15) | 47 | 12.1 |
| **Insomnia** | | |
| Normal (<6) | 205 | 52.8 |
| Insomnia (≥6) | 183 | 47.2 |
| **Loneliness** | | |
| Low (20–34) | 95 | 24.5 |
| Moderate (35–49) | 212 | 54.6 |
| Moderately high (50–64) | 75 | 19.3 |
| High (65–80) | 6 | 1.5 |
| **Dementia** | | |
| No dementia (3–5) | 258 | 66.5 |
| Dementia (0–2) | 130 | 33.5 |
| **Anxiety** | | |
| Minimal (≤6) | 154 | 39.7 |
| Mild (7–9) | 68 | 17.5 |
| Moderate (10–12) | 52 | 13.4 |
| Severe (≥12) | 114 | 29.4 |

Similarly, age, sex, marital status, educational status, pension facility and activity of daily living were significantly associated with dementia (Table 5).

## Discussion

Psychosocial problems such as anxiety, depression, insomnia, loneliness, and dementia are common among older people which are the main causes of impairment, morbidity and mortality in older adults.

In this study, prevalence of depression was 66.8% among older adults where 38.1%, 16.8% and 12.1% were mild, moderate and severe depression respectively. This finding is closely approximates to the previous studies conducted in the community of Kavre [32] and Sunsari and Morang [33]districts of Nepal, which reported 53.1% and 55.8% depression in older adults respectively. Similarly, systematic review reported 25.5% to 60.6% prevalence of depression among older adult in community setting of Nepal [19]. However, our finding is higher than the studies in China [34], Ethiopia [35], Sri Lanka [36], and Pakistan [37], which showed 37.34%, 41.85%, 27.8%, 40.6% depression respectively. It indicates that depression is still high among older adults in our context than the global mean. The discrepancy rates in prevalence of depression between these studies may be the cause of the disparities in regional and geographical characteristics, such as cultural background, social involvement, access to healthcare, and variation in measurement tool. Positive screening followed by clinical diagnosis is ideal; however, the current study did not account for this aspect.

Our study revealed that 60.3% of older adults experienced anxiety symptoms where 17.5% mild, 13.4% moderate and 29.4% severe anxiety. This finding is higher than the studies conducted in Nepal [15], Germany [38], Turkey [39], China [34] and Myanmar [40], which revealed 13.9%, 17.1%, 32.74%, and 39.4% of anxiety respectively. Similarly, anxiety ranged

**Table 3. Association between anxiety and depression and selected variables n = 388.**

| Variables | Anxiety | | | Depression | | |
|---|---|---|---|---|---|---|
| | Absent | Present | P value | Absent | Present | p value |
| **Age groups in years** | | | | | | |
| Young old (65–74) | 90 (37.5) | 150 (62.5) | 0.408 | 80 (33.3) | 160 (66.7) | 0.717 |
| Middle old (75–84) | 54 (44.6) | 67 (55.4) | | 41 (33.9) | 80 (66.1) | |
| Old old (85 and above) | 10 (37.0) | 17 (63.0) | | 7 (25.9) | 20 (74.1) | |
| **Sex** | | | | | | |
| Male | 83 (40.7) | 121(59.3) | 0.673 | 76 (37.3) | 128(62.7) | 0.060 |
| Female | 71 (38.6) | 113 61.4) | | 52 (28.3) | 132(71.7) | |
| **Marital status** | | | | | | |
| Single (widow and widower) | 37 (35.9) | 66 (64.1) | 0.362 | 102 (35.8) | 183 (64.2) | 0.051 |
| Married and living with spouse | 117(41.1) | 168(58.9) | | 26 (25.2) | 77 (74.8) | |
| **Family type** | | | | | | |
| Nuclear | 28 (38.9) | 44 (61.1) | 0.878 | 23 (31.9) | 49 (68.1) | 0.834 |
| Joint | 126 (39.9) | 190 (60.1) | | 105 (33.2) | 211 (66.6) | |
| **Education status** | | | | | | |
| Illiterate | 65 (34.8) | 122 (65.2) | 0.055 | 50 (26.7) | 137 (73.3) | **0.012** |
| Literate | 89 (44.3) | 112 (55.7) | | 78 (38.8) | 123 (61.2) | |
| **Pension facility** | | | | | | |
| No | 125 (38.0) | 204 (62.0) | 0.107 | 98 (29.8) | 231 (70.2) | **0.002** |
| Yes | 29 (49.2) | 30 (50.8) | | 30 (50.8) | 29 (49.2) | |
| **Presence of co-morbidity** | | | | | | |
| No | 54 (41.5) | 76 (58.5) | 0.597 | 50 (38.5) | 80 (61.5) | 0.104 |
| Yes | 100 (38.8) | 158(61.2) | | 78 (30.2) | 180 (69.8) | |
| **Activity of daily living** | | | | | | |
| Impaired (<6) | 8 (32.0) | 17 (68.0) | 0.416 | 5 (20.0) | 20 (80.0) | 0.153 |
| Full functioning (6) | 146 (40.2) | 217 (59.8) | | 123 (33.9) | 240 (66.1) | |
| **Smoking status** | | | | | | |
| Never smoker | 83 (39,5) | 127 (60.5) | 0.813 | 74 (35.2) | 136 (64.8) | 0.276 |
| Past smoker | 50 (38.5) | 80 (61.5) | | 36 (27.7) | 94 (72.3) | |
| Current smoker | 21 (43.8) | 27 (56.2) | | 18 (37.5) | 30 (62.5) | |
| **Alcohol drinker** | | | | | | |
| Abstainer | 139 (39.4) | 214 (60.6) | 0.735 | 117 (33.1) | 236 (66.9) | 0.763 |
| Former drinker | 3 (33.3) | 6 (66.7) | | 2 (22.2) | 7 (77.8) | |
| Current drinker | 12 (46.2) | 14 (53.8) | | 9 (34.6) | 17 (65.4) | |

from 0.2% to 32.2% in older persons in low- and middle-income countries in Africa, Asia, and South America [41]. Another review found that the prevalence of anxiety disorders in late life ranged from 3.2% to 14.2% in Western countries [42].These variations in anxiety prevalence may be attributable to interviewing methodologies, screening tools sensitivity and usage of different cut-off scores to define anxiety.

Insomnia is one of the most frequent sleep disorders among old adult. Nearly half (47.2%) of participants of this study had insomnia. This finding is consistent with the result of the reviewed study, which reported 30 to 48% of prevalence of insomnia among of older people [43]. However; our finding is lower than the studies conducted in Nepal [44, 45], India [46] and Egypt [47] and higher than the study done in Taiwan [48]. These discrepancies may be caused up by inconsistent concept of insomnia and variation in measurement of tool. Additionally, a number of factors, including food, physical activity, lifestyle, and interviewing

**Table 4. Association between insomnia and loneliness and selected variables n = 388.**

| Variables | Insomnia Status | | p value | Loneliness Status | | p value |
|---|---|---|---|---|---|---|
| | **Absent** | **Present** | | **Absent** | **Present** | |
| **Age groups in years** | | | | | | |
| Young old (65–74) | 129 (53.8) | 111 (46.2) | 0.231 | 65 (27.1) | 175 (72.9) | 0.243 |
| Middle old (75–84) | 66 (54.5) | 55 (45.5) | | 26 (21.5) | 95 (78.5) | |
| Old old (85 and above) | 10 (37.0) | 17 (63.0) | | 4 (14.8) | 23 (85.2) | |
| **Sex** | | | | | | |
| Male | 107 (52.5) | 97 (47.5) | 0.873 | 53 (26.0) | 151 (74.0) | 0.471 |
| Female | 98 (53.3) | 86 (46.7) | | 42 (22.8) | 142 (77.2) | |
| **Marital status** | | | | | | |
| Single (widow and widower) | 49 (47.6) | 54 (52.4) | 0.212 | 18 (17.5) | 85 (82.5) | 0.054 |
| Married and living with spouse | 156 (54.7) | 129 (45.3) | | 77 (27.0) | 208 (73.0) | |
| **Family type** | | | | | | |
| Nuclear | 43 (59.7) | 29 (40.3) | 0.195 | 21 (29.2) | 51 (70.8) | 0.306 |
| Joint | 162 (51.3) | 154 (48.7) | | 74 (23.4) | 242 (76.6) | |
| **Education status** | | | | | | |
| Illiterate | 104 (55.6) | 83 (44.4) | 0.290 | 36 (19.3) | 151 (80.7) | **0.021** |
| Literate | 101 (50.2) | 100 (49.8) | | 59 (29.4) | 142 (70.6) | |
| **Pension facility** | | | | | | |
| No | 171 (52.0) | 158 (48.0) | 0.423 | 76 (23.1) | 253 (76.9) | 0.134 |
| Yes | 34 (57.6) | 25 (42.4) | | 19 (32,2) | 40 (67.8) | |
| **Presence of co-morbidity** | | | | | | |
| No | 82 (63.1) | 48 (36.9) | **0.004** | 29 (22.3) | 101 (77.7) | 0.479 |
| Yes | 123 (47.7) | 135 (52.3) | | 66 (25.6) | 192 (74.4) | |
| **Activity of daily living** | | | | | | |
| Impaired (<6) | 4 (16.0) | 21 (84.0) | **<0.001** | 6 (24.0) | 19 (76.0) | 0.954 |
| Full functioning (6) | 201 (55.4) | 162 (44.6) | | 89 (24.5) | 274 (75.5) | |
| **Smoking status** | | | | | | |
| Never smoker | 109 (51.9) | 101 (48.1) | 0.573 | 56 (26.7) | 154 (73.3) | 0.340 |
| Past smoker | 73 (56.2) | 57 (43.8) | | 31 (23.8) | 99 (76.2) | |
| Current smoker | 23 (47.9) | 25 (52.1) | | 8 (16.7) | 40 (83.3) | |
| **Alcohol drinker** | | | | | | |
| Abstainer | 185 (52.4) | 168 (47.6) | 0.586 | 89 (25.2) | 264 (74.8) | 0.507 |
| Former drinker | 4 (44.4) | 5 (55.6) | | 1 (11.1) | 8 (88.9) | |
| Current drinker | 16 (61.5) | 10 (38.5) | | 5 (19.2) | 21 (80.8) | |

techniques, could be responsible for the observed variance in prevalence. Yet, those factors were not taken into consideration in this study.

The study found 33.5% of older persons had dementia. This finding is comparable with a study conducted in Indonesia, which showed dementia among 39.42% of older people in rural areas and 29.15% in suburban areas [49]. Similarly, 36.6% of Iranians had dementia to some extent [50]. However, lower prevalence of dementia was reported in other studies done in Nepal [51–54] and Jordan [55]. Another meta-analysis reported that a substantial percentage of dementia were undiagnosed; more than 60% of those with dementia are not detected in the general population [56].Variability in prevalence estimates of dementia due to geographical variations, variations in survival duration, the distribution of risk/protective variables, access to health care, study design and technique among studies.

**Table 5. Association between dementia and selected variables n = 388.**

| Variables | Dementia Status | | χ2 | p value |
|---|---|---|---|---|
| | **No dementia** | **Dementia** | | |
| **Age groups in years** | | | | |
| Young old (65–74) | 170 (70.8) | 70 (29.2) | 6.174 | **0.046** |
| Middle old (75–84) | 74 (61.2) | 47 (38.8) | | |
| Old old (85 and above) | 14 (15.9) | 13 (48.1) | | |
| **Sex** | | | | |
| Male | 155 (76.0) | 49 (24.0) | 17.373 | **<0.001** |
| Female | 103 (56.0) | 81 (44.0) | | |
| **Marital status** | | | | |
| Single (widow and widower) | 57 (55.3) | 46 (44.7) | 7.832 | **0.005** |
| Married and living with spouse | 201 (70.5) | 84 (29.5) | | |
| **Family type** | | | | |
| Nuclear | 54 (75.0) | 18 (25.0) | 2.870 | 0.090 |
| Joint | 204 (64.6) | 112 (35.4) | | |
| **Education status** | | | | |
| Illiterate | 97 (51.9) | 90 (48.1) | 34.647 | **<0.001** |
| Literate | 161 (80.1) | 40 (19.9) | | |
| **Pension facility** | | | | |
| No | 208 (63.2) | 121 (36.8) | 10.403 | **0.001** |
| Yes | 50 (84.7) | 9 (15.3) | | |
| **Presence of co-morbidity** | | | | |
| No | 85 (65.4) | 45 (34.6) | 0.108 | 0.742 |
| Yes | 173 (67.1) | 85 (32.9) | | |
| **Activity of daily living** | | | | |
| Impaired (<6) | 11 (44.0) | 14 (56.0) | 6.069 | **0.014** |
| Full functioning (6) | 247 (68.0) | 116 (32.0) | | |
| **Smoking status** | | | | |
| Never smoker | 147 (70.0) | 63 (30.0) | 2.636 | 0.268 |
| Past smoker | 82 (63.1) | 48 (36.9) | | |
| Current smoker | 29 (60.4) | 19 (39.6) | | |
| **Alcohol drinker** | | | | |
| Abstainer | 234 (66.3) | 119 (33.7) | 0.095 | 0.953 |
| Former drinker | 6 (66.7) | 3 (33.3) | | |
| Current drinker | 18 (69.2) | 8 (30.8) | | |

Dementia prevalence increased with increasing years of age and decreased with more years of education [57]. This study also found that age, sex, education, marital status, pension and daily living activities were significantly associated with dementia. Similarly, another study conducted in Australia reported that age and functional impairment are related to dementia [58].

Loneliness frequency among community-dwelling older people varies by country to country due to their nature of family structure and support system. Our study showed that the 75.5% older adult experienced moderate to high level of loneliness (moderate-54.6%, moderate to high-19.3% and high-1.5).The finding is quite similar to study conducted in Nepal which revealed 38.7% and 16.9% experienced moderate to severe level of loneliness [59]. Almost similar findings revealed by the study done in Indonesia [60], in which loneliness was 64% among older people. However, studies reported lower prevalence of loneliness in Sweden [61], Norway [62] and China [63]. The difference in findings might be due to breakdown of

conventional family structures, insufficient social welfare, and a lack of mental health care which may contribute to loneliness among seniors.

The current investigation discovered a significant association between depression and pension and education. Being illiterate is more prone to have depression than literate. This evidence agreed with the evidence of Thailand [64], Ethiopia [65] and Ezypt [66]. Similarly, present study found that not receiving pension or financial dependent is associated risk factor. This is similar to study done in Ethiopia which found that seniors with low incomes were approximately two times as likely as those with high incomes to experience depression [67]. Likewise, other studies in Asia (Myanma) [40], North India [68] and Portugal [69] showed similar result. Low-income older persons had a more difficult time accessing health services and care, which may have been linked to higher levels of depression [67].

Insomnia has been linked to a number of chronic disorders such as coronary artery disease, chronic obstructive pulmonary disease, and brain hemorrhage [70, 71]. According to the results of the current study, co-morbidity and daily life activities were significantly associated with insomnia. The older adult who had co-morbid diseases had higher chance for insomnia compared to without co-morbidity. This finding is agreed with other studies done in Northern Taiwan [72] and Nepal [44]. Additionally; it has been shown that older adults' sleep quality is negatively impacted by drugs [44].

The findings of the study provide information on prevalence of various psychosocial problems among older adult which might be useful to health care provider for planning and implementation of preventive and curative strategies on psychosocial problems for older people. Despite of this, the study has some limitations. This is a cross-sectional study, it cannot definitively any causal pathway; it can only determine the links between psychosocial problems and the associated factors. All the psychosocial assessment tools were self -reported tool which is subjected to recall bias. This study is conducted in community dwelling older adults' psychosocial problems, so results cannot be generalized to older people residing in geriatric home and health care settings.

## Conclusion

This study concluded that psychosocial problems such as anxiety, depression, loneliness, insomnia and dementia are common among older people. Among them, higher proportions suffer from depression, loneliness and anxiety. Age, functional dependency, sex, co-morbidity, financial dependence, education, and occupation are risk factors for psychosocial issues in older people. Therefore, local health care provider must organize and carry out routine screening, counselling, and awareness programs for the older adults in the community. Further, there is need to develop and implement health care strategy by local health care planner to prevent, treat and manage the psychosocial problems among this risk groups.

## Acknowledgments

Researchers would like to thank Management Committee of Chitwan Medical College Teaching Hospital. Last but not least, researchers heartfelt thank go to participants who gave their valuable information and time for this study.

## Author Contributions

**Conceptualization:** Sunita Poudyal, Kalpana Sharma, Hem Kumari Subba, Ramesh Subba.

**Data curation:** Kalpana Sharma, Hem Kumari Subba.

**Formal analysis:** Sunita Poudyal, Kalpana Sharma.

**Funding acquisition:** Sunita Poudyal, Kalpana Sharma.

**Methodology:** Sunita Poudyal, Kalpana Sharma, Hem Kumari Subba, Ramesh Subba.

**Project administration:** Sunita Poudyal, Kalpana Sharma.

**Resources:** Sunita Poudyal, Hem Kumari Subba.

**Software:** Sunita Poudyal, Kalpana Sharma.

**Supervision:** Sunita Poudyal, Kalpana Sharma, Hem Kumari Subba.

**Validation:** Sunita Poudyal, Kalpana Sharma.

**Writing – original draft:** Sunita Poudyal, Kalpana Sharma, Hem Kumari Subba, Ramesh Subba.

**Writing – review & editing:** Sunita Poudyal, Kalpana Sharma, Hem Kumari Subba, Ramesh Subba.

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
