## [Decision Letter · Decision Letter 0]

15 Jul 2024

PONE-D-24-12655Psycho-Social Problems among Older People Residing in Chitwan, Nepal - A Cross-sectional StudyPLOS ONE

Dear Dr. Poudyal,

Thank you for submitting your manuscript to PLOS ONE. After careful consideration, we feel that it has merit but does not fully meet PLOS ONE’s publication criteria as it currently stands. Therefore, we invite you to submit a revised version of the manuscript that addresses the points raised during the review process.

We look forward to receiving your revised manuscript.

Kind regards,

Kamlesh Kumar Sahu

Academic Editor

PLOS ONE

Journal Requirements:

"Nepal Health Research Council as provincial research grant."

**Additional Editor Comments:**

these point must be addressed:

community will be better than writing a city name in objective

refrences need to recheck and correct as per the journal stlye

conclusion needs to be in a separate heading

term "elderly" as well as being ageist terms are not inclusive terms therefore use the term older people/persons.

Reviewers' comments:

Reviewer's Responses to Questions

**Comments to the Author**

1. Is the manuscript technically sound, and do the data support the conclusions?

Reviewer #1: Partly

Reviewer #2: Yes

Reviewer #3: Yes

2. Has the statistical analysis been performed appropriately and rigorously? 

Reviewer #1: No

Reviewer #2: Yes

Reviewer #3: Yes

3. Have the authors made all data underlying the findings in their manuscript fully available?

Reviewer #1: No

Reviewer #2: Yes

Reviewer #3: Yes

4. Is the manuscript presented in an intelligible fashion and written in standard English?

Reviewer #1: Yes

Reviewer #2: Yes

Reviewer #3: Yes

5. Review Comments to the Author

Reviewer #1: The manuscript has the potential to show ageing in Nepal, considering that there is little scientific evidence on the ageing process in some Asian countries. However, its treatment is purely descriptive and this reduces the scientific value of the proposal, as a frequency and/or prevalence, manuscript is more frequent in reports and other types of documents. Therefore, the proposal requires rethinking the data analyses and moving them to a multivariate level, but in addition to this substantive change to the methodology, the following is suggested to the authors:

(1) Avoid the term "elderly" as well as being ageist, it is an under-inclusive term and therefore use the term older people/persons.

(2) The introduction is also loaded with an ageist and negative view of old age, as a process of decline, multimorbidity, etc. Population ageing is a success story of societies and the group of older people is highly diverse and has different trajectories/ways of ageing.

(3) The methodology should consider whether the scales have been previously validated in older people in Nepal. The proposed descriptive analysis can be included in a first phase and then a mutivariate analysis should be carried out.

Reviewer #2: Refrences need through check as per the journal style there few mistakes

Conclusion need to be in saparete heading

In the objective 'This study aimed to find out the psycho-social problems among older people residing in

Bharatpur, Chitwan.' in place of Bharatpur, Chitwan you may write community

Reviewer #3: community will be better than writing a city name in objective

refrences need to recheck and correct as per the journal stlye

conclusion needs to be in a separate heading

term "elderly" as well as being ageist terms are not inclusive terms therefore use the term older people/persons.

6. PLOS authors have the option to publish the peer review history of their article (what does this mean?). If published, this will include your full peer review and any attached files.

Reviewer #1: No

Reviewer #2: No

Reviewer #3: No

---

## [Author Response · Author response to Decision Letter 0]

3 Sep 2024

Respected editors/reviewers

Thank you very much for your suggestion. We tried to incorporate them in our revise manuscript. Hope to get positive response soon.

Thank you

---

## [Editor Report · Decision Letter 1]

5 Sep 2024

Psycho-social problems among older people residing in community of Chitwan, Nepal - A cross-sectional study

PONE-D-24-12655R1

Dear Dr. Poudyal,

We’re pleased to inform you that your manuscript has been judged scientifically suitable for publication and will be formally accepted for publication once it meets all outstanding technical requirements.

Kind regards,

Kamlesh Kumar Sahu

Academic Editor

PLOS ONE
---

## [Editor Report · Acceptance letter]

9 Sep 2024

PONE-D-24-12655R1 

PLOS ONE

Dear Dr. Poudyal, 

I'm pleased to inform you that your manuscript has been deemed suitable for publication in PLOS ONE. Congratulations! Your manuscript is now being handed over to our production team.

Kind regards, 

on behalf of

Dr. Kamlesh Kumar Sahu 

Academic Editor

PLOS ONE